# InheritSumm: A General, Versatile and Compact Summarizer by Distilling from GPT

**Yichong Xu, Ruochen Xu, Dan Iter, Yang Liu, Shuohang Wang**

**Chenguang Zhu, Michael Zeng**
Microsoft Azure AI
{yicxu,ruox,iterdan,yaliu10,shuowa,chezhu,nzeng}@microsoft.com

## Abstract

While large models such as GPT-3 demonstrate exceptional performance in zeroshot and fewshot summarization tasks, their extensive serving and fine-tuning costs hinder their utilization in various applications. Conversely, previous studies have found that although automatic metrics tend to favor smaller fine-tuned models, the quality of the summaries they generate is inferior to that of larger models like GPT-3 when assessed by human evaluators. To address this issue, we propose INHERITSUMM, a versatile and compact summarization model derived from GPT-3.5 through distillation. INHERIT-SUMM not only exhibits comparable zeroshot and fewshot summarization capabilities to GPT-3.5 but is also sufficiently compact for fine-tuning purposes. Experimental results demonstrate that INHERITSUMM achieves similar or superior performance to GPT-3.5 in zeroshot and fewshot settings. Furthermore, it outperforms the previously established best small models in both prefix-tuning and full-data fine-tuning scenarios.

## 1   Introduction

Recently, the development of large language models (LLMs) like GPT-3 (Brown et al., 2020b) and PaLM (Chowdhery et al., 2022) has largely revolutionized the text summarization community, bringing a new paradigm in the the way summaries are generated. LLMs have demonstrated unparalleled ability to produce highly readable summaries while requiring little to no training data, overwhelmingly prefered by human annotators than those from smaller models (Goyal et al., 2022). Human evaluators note that GPT-3 generates summaries with superior readability and coherence, sometimes even more favored than human-generated summaries (Liang et al., 2022). One of the key advantages of LLMs is their capacity for zeroshot and fewshot learning (Brown et al., 2020b), which enables them to adapt to new domains with ease. There-

fore, LLMs are highly attractive for a wide range of summarization applications.

Despite these remarkable achievements, training and deploying LLMs for summarization is computationally expensive. Deploying an LLM for inference is already impractical for many NLP practitioners, especially for summarization where long input length of the document and demonstrative examples are typical. Moreover, the prohibitive cost to finetune an LLM makes them hard to adapt to various custom domains, when abundant training data is available. This largely hurts the widespread adoption of LLMs for summarization. To this end, Zhu and Zeng (2022) proposes the target of achieving all three aspects of the impossible triangle for NLP models: moderate model size, superior zeroshot / fewshot learning capability, and superior supervised learning capability.

In light of these challenges, we propose INHER-ITSUMM. The model exhibits a compact size and offers cost-effective fine-tuning and deployment capabilities. Its versatility is demonstrated by its applicability in both zero-shot/few-shot settings as well as supervised settings. Furthermore, the model's generalization capability is evidenced by its promising performance across a wide range of benchmarks in all mentioned settings.

INHERITSUMM is trained using knowledge distillation from the GPT-3 model, by mimicking the GPT-generated summaries on general documents. To facilitate this, we curated the GPT-SUMM dataset, with over 7 million document-summary pairs. The documents are collected from general language corpora and GPT-3 generates the corresponding summary for them. Utilizing this dataset, we train a ZCode++ (He et al., 2022) model by following the GPT-generated summaries in both zeroshot and fewshot learning settings.

One important limitation of fewshot learning for summarization is the input length limit: The in-context demonstrations consume valuable input

space, making it challenging to incorporate many in-context examples within the prompt. To address this issue, we propose utilizing *succinct demonstrations* that aim to shorten the demonstration documents and subsequently allow for the inclusion of more in-context examples.

By training on the knowledge in GPTSUMM and using succinct demonstrations, we show that INHERITSUMM is able to closely match, or sometimes even outperform GPT-3 in zeroshot and fewshot summarization. Therefore, our INHERITSUMM with only 398M parameters is able to achieve the impossible triangle (Zhu and Zeng, 2022) for the summarization task. We show that INHERITSUMM has strong performance and most versatile summarization capability to date: it achieves best or close-to-best performance in all the settings including zeroshot/fewshot learning with prompts, fewshot learning with prefix tuning, as well as fully supervised learning.

Our key contribution are three-fold. Firstly, we build the GPTSUMM dataset with more than 4M paragraph-summary pairs by using querying GPT-3.5. To the best of our knowledge, GPTSUMM is the largest corpus to date focusing on distilling from GPT-3.5, and is the first one focusing on summarization. Secondly, we build INHERITSUMM based on the GPTSUMM corpus. We show that INHERITSUMM exhibits versatile capabilities across zeroshot, fewshot, and supervised settings, achieving the impossible triangle for summarization (Zhu and Zeng, 2022). Lastly, we propose new methods to include in-context examples for fewshot summarization that improves the performance.

## 2 Problem Statement and Overview

In summarization, the goal is to summarize the input document $D$ into a short summary $Y$. For zeroshot and fewshot learning, we can add a prompt $P$ to the document to familiarize the model (parameterized with $\theta$) with the given task. A sequence-to-sequence model then takes the input $X = [P; D]$ [1] and predicts the summary $\hat{Y}$.

In zeroshot learning, $P$ is a short description of the task, e.g., "summarize the following document into a short paragraph". For fewshot learning with in-context examples, the prompt consists of several descriptive examples along with an instruction, i.e., $P = [X_1; Y_1, ..., X_n; Y_n; I]$, where $X_i$ and $Y_i$ are

| Datasets | Domain | # Docs (%) | Length |
|---|---|---|---|
| The Pile | General | 5.3M | 1,296 |
| ArXiv | Academic | 203k | 6039 |
| CNN/DM | News | 287k | 781 |
| WikiHow | Instructions | 230k | 578 |

Table 1: Sources of Documents in GPTSUMM.

illustrative examples, and $I$ is the task description. For prefix tuning and supervised learning, $P$ is empty and $X = D$. We tune parameters $\theta$ to adapt the model to a given task.

Figure 1 describes our overall method. We distill summarization skills from GPT-3 by using it to generate summaries for documents from different domains, including general documents used for language modeling, as well as specialized documents from labeled datasets. The GPT-generated summaries and (relatively fewer) human generated summaries from labeled datasets forms our GPTSUMM dataset. We then use GPTSUMM to train a seq2seq model, and then adapt it to zeroshot/fewshot learning, prefix tuning and supervised learning summarization.

In the following subsections, we first describe the method to build the GPTSUMM dataset in Section 3, and then introduce the model training process in Section 4. Finally we describe our method to adapt the pretrained INHERITSUMM model to zeroshot, fewshot, and supervised summarization.

## 3 Distillation Data Collection of GPTSUMM

We discuss the construction of GPTSUMM as the data for distillation in this section.

### 3.1 Document Collection

To increase the generalization of downstream models, we collect a corpus of documents from various sources. We first include 3.1M documents from the Pile corpus (Gao et al., 2020) We filter out non-English documents or the ones with too many non-character symbols [2]. We truncate each document to be within 4096 words and remove duplicates following (Smith et al., 2022). To include document from diverse domains and get closer to downstream tasks, we also include documents from arXiv(Cohan et al., 2018), CNN/Daily Mail(See et al., 2017) and WikiHow(Koupaee and Wang,

---

[1] Our method also applies to the case where the prompt content appears after the document $D$.

[2] Namely, we remove documents where the percentage of non-English character symbols is larger than 70%.

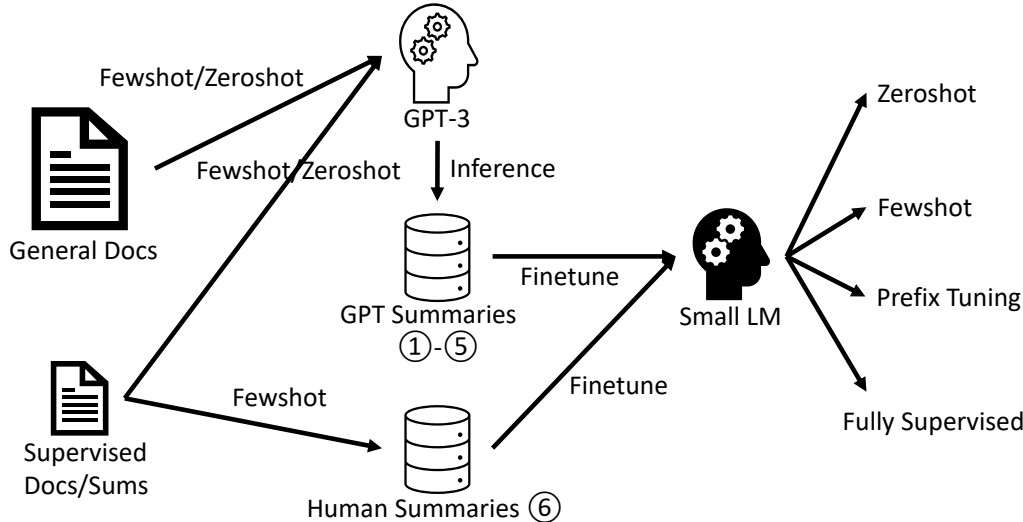

Figure 1: Overview of our method.

| Index | ICD Docs | ICD Summaries | ICD Num | Input Docs | Target Summaries | Quantity |
|---|---|---|---|---|---|---|
| ① | None | None | 0 | General | GPT (zeroshot) | 0.5M |
| ② | General | GPT (from ①) | 1 | General | GPT (fewshot) | 2.6M |
| ③ | Supervised | Supervised | 1 | General | GPT (fewshot) | 2.2M |
| ④ | None | None | 0 | Supervised | GPT (zeroshot) | 0.6M |
| ⑤ | Supervised | Supervised | up to 4 | Supervised | GPT (fewshot) | 0.5M |
| ⑥ | Supervised | Supervised | up to 4 | Supervised | Supervised | 0.6M |

Table 2: Modes of generation in GPTSUMM. ICD means in-context demonstrations. General means documents from the Pile corpus. Supervised corresponds to the labeled datasets in Table 1. The quantity listed are after the filtering steps specified in Sec 3.

2018) datasets. Table 1 describes the composition of documents in detail.

## 3.2 Summary Generation

We utilize the GPT-3.5 model, specifically the `text-davinci-002` variant, to generate summaries. To adapt to downstream use cases, we apply different inputs, prompts, number of in-context demonstrations and zeroshot / fewshot settings to GPT to continue improving the quality of generated data, as shown in Table 2.

Initially, we collect instructions from the PromptSource dataset (Bach et al., 2022) and filter out those that only produce a "subject line" rather than a complete summary. This process yields a final set of 24 instructions. To generate examples for fewshot learning, we use these instructions to produce summaries for 500k documents in the Pile corpus (referred to as "General") in a zeroshot manner, resulting in data type ①. After removing summaries with low quality (as detailed in section 3.3), we use these summaries as in-context examples to gener-

ate summaries for an additional 2.6M documents in the Pile, which we denote as data type ②. In addition to the zeroshot examples from GPT, we also leverage document-summary pairs from supervised datasets (Table 1) as demonstrations to obtain data type ③.

For parts ④⑤⑥, we use the supervised datasets as the input documents. In ④ and ⑤, we utilize GPT to generate the summaries in either zeroshot or fewshot settings, using the supervised datasets as in-context demonstrations. Finally, in ⑥, we employ the supervised datasets themselves to learn in a multitask learning fashion. We use a specific "following" instruction (detailed in Appendix A) to follow the in-context demonstrations for part ⑥ to enable the model to produce diverse outputs compared to ⑤. Through this approach, our model can follow the given instructions to generate distinct summaries for the same documents.

### 3.3 Quality Assessment

To ensure the quality and relevance of the generated summaries, we estimate the summary quality and apply a filtering process consisting of multiple stages. The first step involves retaining only those summaries with a "finish reason" of "stop." This indicates that the model has reached a natural stopping point, which serves as a reliable signal for the summary's quality.

Subsequently, the text undergoes post-processing to eliminate any superfluous content. If the text contains a string that indicates another round of generation, such as consecutive newlines, the text is divided at the consecutive newlines, and only the first part is retained.

We then remove summaries whose word count is less than 10, greater than 512, or exceeds the length of the document.

Following this, the generated summary is assessed using the ROUGE metric to gauge its overlap with the original document. In this case, we treat the original document as the ground truth and the produced summary as the prediction. As a high-quality summary is expected to encompass a subset of the document's information, the recall score largely depends on the document's length and is not particularly informative for evaluating summary quality. Consequently, we rely solely on precision scores for filtering. Specifically, we retain the summary only if its ROUGE-1 precision with the document exceeds 0.6 and its ROUGE-2 precision falls within the range of 0.25 to 0.8. We establish an upper bound for ROUGE-2 precision because we observed that the GPT model occasionally copies sentences from the original document, which may include excessive, irrelevant details.

The filtering process is crucial for ensuring that the generated summary is of high quality and pertinent to the input document. Filtering is particularly important during the bootstrapping generation of data type ①, where we filtered out 17% of the produced summaries. The filtering rate is significantly lower for other data types. For example, less than 5% of the summaries were filtered for data type ②, where one-shot in-context demonstration was applied.

### 3.4 Succinct Demonstrations

Prior work (Brown et al., 2020a) has discovered that an increasing number of in-context examples can enhance the overall quality of generation. How-ever, documents in the summarization task are typically lengthy, leading to a reduced usable input length for GPT and distilled models when including numerous in-context examples. To alleviate this problem, we propose to truncate the document and summary to include more examples. To do this, we first truncate all training documents and summaries in the supervised datasets to length $k$ ($k = 256$ in our experiments). For every truncated document/summary, we add a suffix "<omitted, $l$ words in total>" to the text, where $l$ is the length of the original text. More specifically, for every document $D$ longer than $k$ words, we construct a succinct truncated document $D' = [d_1, d_2, ..., d_k, \text{<omitted}, l(D) \text{ words in total>}]$, where $d_j$ is the $j$-th word in $D$, and similarly for every summary $S$. We then add up to $M$ ($M = 4$ in our experiments) succinct document-summary pairs as in-context examples for any given input document, and ask the model (either GPT or distilled models) to follow the in-context demonstrations. We apply succinct demonstrations to data ⑤ and ⑥ with ICDs from supervised datasets. We do not apply succinct demonstrations to ② and ③ since we observe that altering the number of in-context examples does not significantly impact the quality of generated summaries for general documents from the Pile corpus.

## 4 Model Training

We proceed to train a seq2seq transformer language model ZCode++ (He et al., 2022) parameterized by $\theta$, utilizing the GPTSUMM data. The objective is to minimize negative log-likelihood given the inputs with prompts $[P; X]$:

$$L(\theta) = \sum_{i=1}^{|Y|} \log \mathbb{P}_\theta(y_i | P, X, y_1, ..., y_{i-1}) \quad (1)$$

where $\mathbb{P}_\theta$ is the probability distribution given by model with parameters $\theta$. We train the model in a multi-task manner, where we mix all the data from ① to ⑥ and sample from the data pool in every minibatch. To better balance the model towards general applications, we adjust the sampling ratio between each task to up-sample smaller data modes and down-sample larger ones. Analogous to the data generation in GPTSUMM, we also truncate the in-context examples in data ⑤ and ⑥ as described in Section 3.4.

**In-Context Demonstrations in Distillation.** To improve the GPT generation quality, most of the

data in GPTSUMM are generated in a fewshot manner. A natural way to train the downstream model is to keep the input same as GPT. We call the corresponding model INHERITSUMM-Consistent. However, the resulting model might not be good at zeroshot summarization since most data are in fewshot format. To obtain a model good at both zeroshot and fewshot summarization, we propose to randomly convert some of the fewshot examples to zeroshot examples. More specifically, we train another model where we randomly include 0 to 4 in-context examples by excluding examples in the prompts. We use zeroshot learning for data ① and ④, 0 or 1 examples for ② and ③, and 0 to 4 examples for ⑤. We call the corresponding model INHERITSUMM-Balanced. Note that data ⑥ does not involve GPT generation, and we always include 1 to 4 in-context examples in our training.

### 4.1 Adapting to Application Scenarios

After training the INHERITSUMM model on GPT-SUMM, we adapt the INHERITSUMM to three different summarization settings.

**Zeroshot and fewshot learning with prompts.** The ability to adapt to different tasks with no or few examples is one of the most important characteristics of large models like GPT. We test the pretrained INHERITSUMM model in exactly the same way as GPT. For zeroshot learning, we randomly sample one instruction from PromptSource (Bach et al., 2022). For fewshot learning, we include up to 4 examples (decided by the input document length) with succinct demonstrations from the corresponding training set.

**Fewshot learning with prefix tuning.** We follow the setting in Chen and Shuai (2021) and Chen et al. (2022) to prefix-tune our model. For every task $t$, we add $K$ task-specific prefix vectors at every layer of the encoder and decoder for INHERITSUMM, parameterized by $\theta^t$. Then we freeze all other parameters in INHERITSUMM and tune $\theta^t$ by minimizing (1).

**Fully Supervised Learning.** In this setting we use all the data in downstream datasets to perform fully supervised finetuning. All the model parameters are finetuned by minimizing loss (1), without any prompts or instructions.

## 5 Experiments

We detail the experiment results in this section. We first specify the implementation details and

hyperparameters in Sec. 5.1, and then introduce the main experiment results in Sec. 5.2.

### 5.1 Implementation Details

| Hyper-parameter | Value |
|---|---|
| Warmup Steps | 10,000 |
| Learning Rates | 2e-5 |
| Batch Size | 144 (base), 120 (large) |
| Local Attention Window | 256 |
| Global Layers | 6/11 (base), 11/23 (large) |
| Weight Decay | 0.01 |
| Training Steps | 300k |
| Learning Rate Decay | Linear |
| Adam $\epsilon$ | 1e-6 |
| Adam $\beta_1$ | 0.9 |
| Adam $\beta_2$ | 0.999 |
| Gradient Clipping | 1.0 |
| Beam search size | 5 |

Table 3: Hyper-parameters for Training INHERIT-SUMM.

We use the ZCode++ model (He et al., 2022) as the language model backbone for training INHERIT-SUMM. We choose ZCode++ over other pretrained models since it achieves state-of-art performance on summarization tasks after finetuning. We experimented with both the pretrained Z-Code++ base and Z-Code++ large model. We train the model on the GPTSUMM corpus with seq2seq loss (1) for 300K steps, with a maximum input length of 3072. We summarize the hyperparameters in Table 3. To reduce the memory usage we follow ZCode++ to use Fusion-in-Encoder. We use two layers as the global attention layer and local layers have a window size of 256.

**Test settings.** We test INHERITSUMM in 4 different settings: i) zeroshot learning with random instructions from PromptSource (Bach et al., 2022), ii) 4-shot learning with instructions, iii) 10-shot learning with prefix tuning, and iv) fully supervised learning. i) and ii) are prompt-based settings typically employed by large models like GPT, whereas iii) and iv) are more traditional settings for smaller models.

**Baselines.** We compare with GPT `text-davinci-002`, the original ZCode++, BART (Lewis et al., 2019) and UniSumm (Chen et al., 2022). Due to hardware and compute limits, we do not compare with GPT in prefix tuning and fully supervised settings. For fewshot learning with prefix tuning, We follow Li and Liang (2021) to tune prefix embeddings at every encoder and de-

| Datasets | Domain | Avg. D/S Length |
|----------|--------|-----------------|
| MultiNews | News | 1,979 / 275 |
| XWiki | Wikipedia | 971 / 85 |
| SAMSum | Dialogue | 136 / 24 |
| Reddit-TIFU | Forum | 496 / 29 |
| BigPatent | Legal | 2,853 / 119 |

Table 4: Statistics of testing datasets. Avg. D/S Length is the average number of GPT tokens for document/summary for the corresponding test set.

coder layer. For local layers in Fusion-in-Encoder of ZCode++ and INHERITSUMM, one set of prefix embeddings are inserted at every local window for every local layer.

**Testing Datasets.** We test on 5 summarization datasets on diverse domains (a summary in Table 4):

**MultiNews** (Fabbri et al., 2019) is a multi-document news summarization dataset with news from different sources.

**XWiki** (Perez-Beltrachini and Lapata, 2021) is a cross-lingual summarization dataset focusing on Wikipedia articles. We use the English data with paired documents and summaries.

**SAMSum** (Gliwa et al., 2019) is a dialogue summarization with chit-chat dialogues in online chatting styles like Messenger and WhatsApp. Both the dialogue and summary are human-written by expert linguists.

**Reddit-TIFU** (Kim et al., 2019) is another dialogue summarization dataset focusing on the online forum Reddit. The language style on Reddit is significantly different from news articles.

**BigPatent** (Sharma et al., 2019) is a legal document summarization dataset with US patent documents and human-written abstracts. Documents come from 9 different domains.

Some of the datasets have an extensive test set that takes a long time to test GPT on. Also, the GPT API has a content filter that rejects some of the input documents. Therefore, we randomly sample 500 documents from the test sets, and choose those that pass GPT's content filter to compare the baselines.

### 5.2 Experiment Results

We summarize our main results for the four settings in Table 5. For simplicity and space reasons, we include ROUGE-2 scores in our experiment results.

We choose ROUGE-2 as it has a better correlation with human judgments on meta-eval benchmarks (Zhong et al., 2022). All the experiment results are from our own runs.

**Zeroshot learning.** INHERITSUMM models demonstrated better performance than all the baselines in 3 out of 5 test datasets, as well as on the average over the 5 datasets. INHERITSUMM's performance is inferior to BART or UniSumm on MultiNews and Xwiki respectively, possibly because the summary on these two datasets are longer than the other datasets (this is also true for GPT-3.5 model). INHERITSUMM gets higher performance than the teacher GPT-3.5 model. This is probably because INHERITSUMM is specialized in summarization, while GPT-3.5 might fail to follow the instructions to summarize the input document.

Among the four variants of INHERITSUMM, the base INHERITSUMM-Balancedachieves the highest ROUGE score. The models trained in the balanced way receive more zeroshot examples in its training process, which probably makes them better at zeroshot learning. However, this is not true for INHERITSUMM-Large models, where the balanced model is slightly behind the consistent model. This might be because large models are more capable when generalizing across different settings, and the data composition (whether balanced or consistent) is not very important for large models.

**Fewshot learning with instructions.** GPT-3.5 achieves the best average ROUGE score of 12.27 in this setting, whereas our INHERITSUMM models are only behind GPT-3.5 by a small gap. Among the four variants, INHERITSUMM-Balanced-Large achieves the best score of 12.15, slightly behind GPT-3.5. INHERITSUMM-Balanced-Large also beats GPT-3.5 on 2 of the test datasets. Large models are generally better in fewshot learning than base models. The performance between models trained with balanced or consistent data is comparable, likely because both models receive large quantities of fewshot data in their training.

**Fewshot learning with prefix tuning.** INHERITSUMM generally achieves the best performance in this setting, only loses to UniSumm on Bigpatent by a small margin. INHERITSUMM-Consistent-base is the best in the average performance for the prefix tuning setting. The prefix tuning results of INHERITSUMM are also significantly better than the original ZCode++ models, suggesting the effectiveness of our distillation training.

| Task | Datasets | Z-Base | Z-Large | GPT-3.5 | BART | UniSumm | IS-Base-B | IS-Base-C | IS-Large-B | IS-Large-C |
|---|---|---|---|---|---|---|---|---|---|---|
| Zeroshot | Samsum | 0.78 | 0.01 | 10.85 | 8.05 | 4.58 | **17.82** | 16.13 | 15.05 | 16.73 |
| | Xwiki | 1.41 | 0.12 | 6.97 | 5.47 | **9.31** | 8.35 | 8.02 | 8.50 | 8.59 |
| | Reddit Tifu | 0.09 | 0.00 | 4.02 | 3.03 | 4.41 | 5.75 | 4.94 | **5.97** | 5.93 |
| | Bigpatent | 1.18 | 0.01 | 10.09 | 9.08 | 10.83 | 12.36 | 12.29 | **12.44** | 11.93 |
| | MultiNews | 1.54 | 0.07 | 8.23 | **10.28** | 3.47 | 8.65 | 8.31 | 7.01 | 7.72 |
| | Avg | 1.00 | 0.04 | 8.03 | 7.18 | 6.52 | **10.59** | 9.94 | 9.80 | 10.18 |
| Fewshot(instruct) | Samsum | 0.00 | 0.05 | 18.67 | 1.08 | 0.60 | 18.90 | **18.93** | 18.71 | 17.59 |
| | Xwiki | 0.98 | 0.10 | **11.83** | 1.53 | 3.78 | 9.55 | 9.84 | 9.58 | 9.99 |
| | Reddit Tifu | 0.03 | 0.03 | 6.68 | 0.62 | 0.91 | 7.28 | 7.14 | **8.51** | 7.93 |
| | Bigpatent | 1.22 | 0.01 | 12.85 | 3.55 | 4.45 | 9.98 | 12.71 | **13.18** | 12.88 |
| | MultiNews | 0.92 | 0.07 | **11.32** | 1.92 | 1.33 | 10.21 | 9.76 | 10.77 | 10.04 |
| | Avg | 0.63 | 0.05 | **12.27** | 1.74 | 2.21 | 11.19 | 11.67 | 12.15 | 11.68 |
| Fewshot(Prefix) | Samsum | 14.25 | 15.79 | N/A | 9.88 | 11.37 | 20.97 | **20.99** | 19.49 | 19.89 |
| | Xwiki | 10.47 | 12.07 | N/A | 11.08 | 8.29 | 11.91 | 12.07 | 12.20 | **12.46** |
| | Reddit Tifu | 3.92 | 3.94 | N/A | 2.78 | 6.19 | 6.37 | 6.54 | **6.92** | 5.26 |
| | Bigpatent | 7.13 | 5.86 | N/A | 6.99 | **13.12** | 12.23 | 12.65 | 12.84 | 12.23 |
| | MultiNews | 4.88 | 8.92 | N/A | 11.63 | 10.84 | 11.69 | **12.06** | 10.91 | 11.33 |
| | Avg | 8.13 | 9.32 | N/A | 8.47 | 9.96 | 12.63 | **12.86** | 12.47 | 12.23 |
| Supervised | Samsum | 27.49 | 27.91 | N/A | 29.26 | 22.36 | **30.12** | 29.87 | 28.52 | 28.60 |
| | Xwiki | 21.93 | 21.77 | N/A | 20.21 | 18.05 | 21.70 | 21.74 | **22.48** | 20.63 |
| | Reddit Tifu | 10.66 | 10.37 | N/A | 11.33 | 8.42 | 11.57 | **11.74** | 10.25 | 10.31 |
| | Bigpatent | 17.94 | 12.64 | N/A | 17.88 | 17.38 | 17.99 | 18.03 | 21.67 | **23.11** |
| | MultiNews | 17.66 | 19.11 | N/A | 17.87 | 18.69 | 19.83 | 20.38 | 19.03 | **20.58** |
| | Avg | 19.14 | 18.36 | N/A | 19.31 | 16.98 | 20.24 | 20.35 | 20.39 | **20.65** |
| Mean | Samsum | 10.63 | 10.94 | N/A | 12.07 | 9.73 | **21.95** | 21.48 | 20.44 | 20.70 |
| | Xwiki | 8.70 | 8.52 | N/A | 9.57 | 9.86 | 12.88 | 12.92 | **13.19** | 12.92 |
| | Reddit Tifu | 3.67 | 3.58 | N/A | 4.44 | 4.98 | 7.74 | 7.59 | **7.91** | 7.36 |
| | Bigpatent | 6.87 | 4.63 | N/A | 9.38 | 11.45 | 13.14 | 13.92 | 15.03 | **15.04** |
| | MultiNews | 6.25 | 7.04 | N/A | 10.43 | 8.58 | 12.59 | **12.63** | 11.93 | 12.42 |
| | Avg | 7.23 | 6.94 | N/A | 9.18 | 8.92 | 13.66 | **13.71** | 13.70 | 13.69 |

Table 5: Main Experiment Results. For simplicity, we only include ROUGE-2 scores in the table. IS stands for INHERITSUMM, B stands for balanced and C stands for consistent. The largest and second-largest number in each row are in **bold** and underlined respectively. The "Mean" section at the bottom of the table is the mean performance over 4 different settings.

**Fully supervised learning.** Lastly, INHERIT-SUMM outperforms all the bselines in the fully supervised learning setting as well. INHERITSUMM outperforms the original ZCode++ model, showing the transfer ability of our distillation training. Among the four variant of INHERITSUMM, INHERITSUMM-Consistent-Large gives the best performance. This is likely because large models are more powerful with fully supervised data, and consistent data training is better for the transfer of knowledge.

For average over the 4 settings, INHERITSUMM strongly outperforms all the baselines on the aggregate score over 4 settings, showing that INHERITSUMM is the most versatile model across different training scenarios. The average performance of the 4 variants is quite close.

| Datasets | R-2(zeroshot) | R-2 (fewshot) |
|---|---|---|
| ① + ② | 9.82 | 10.69 |
| ③ | 10.02 | 10.04 |
| ④ | 10.12 | 3.06 |
| ⑤ | 8.80 | 9.95 |
| ⑥ | 6.53 | 6.17 |
| All | 10.59 | 11.19 |

Table 6: Performance of single-dataset training. R-2 stands for ROUGE-2 scores. All scores are averaging over 5 test sets.

### 5.2.1 Analysis

**Effect of different training datasets.** One natural question is about the effect of each part of GPT-SUMM in INHERITSUMM's performance. While it is not possible to test every combination of ①-⑥, we follow the FLAN paper (Wei et al., 2021)

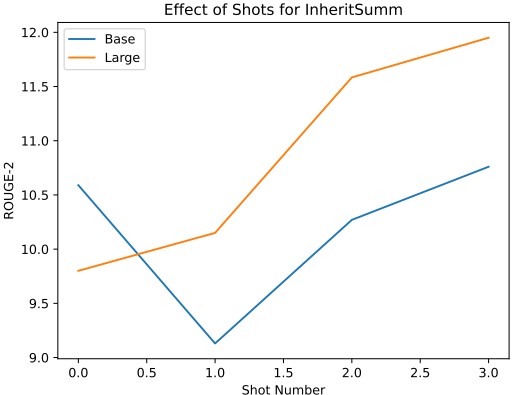

Figure 2: Effect of number of shots for INHERITSUMM base and large model.

to test the performance of INHERITSUMM under individual parts. In Table 6, we train a base INHERITSUMM in the balanced setting with the same hyperparameters on ①+②, ③, ④, ⑤, ⑥[3] respectively. The results show that all the GPT-generated data (① - ⑤) gives better zeroshot/fewshot performance than supervised datasets (⑥), except for ④ on fewshot: this is expected because ④ contains only zeroshot training data. As all parts of data can help boost the zeroshot/fewshot performance, we include all of them as our training data. **Effect of succinct demonstrations.** In order to test the effect of succinct demonstrations, we test INHERITSUMM-Balanced's performance with different number of shots. In Figure 2, we plot the performance of base and large INHERITSUMM-Balancedfrom 0-shot to 4-shots. For both models, the performance improves from 1-shot to 4-shots. For the large model, the performance also improve when we go from zero-shot to 1-shot, but this is not the case for base model. This shows that using the traditional one-shot method may even hurt the performance, possibly due to model capacity reasons. Our succinct prompt method can always incorporate more in-context examples and improve the model's performance.

## 6 Related Works

**Text Summarization** has been extensively explored by the community (Nenkova and McKeown, 2012). Previous works mostly focus on one or two settings of zeroshot/fewshot/supervised learning only. For example, BART (Lewis et al., 2019)

and ZCode++ (He et al., 2022) focuses on supervised summarization. PEGASUS (Zhang et al., 2020) proposes pretraining methods for unsupervised summarization. UniSumm (Chen et al., 2022) explores fewshot summarization with BART-Large. (Goyal et al., 2022) explores zeroshot and fewshot summarization for GPT-3. To the best of our knowledge, we are the first paper to explore the generalization over zeroshot, fewshot, and supervised summarization.

**Model Distillation from GPT** There has been several works distilling knowledge from the GPT series of models. Wang et al. (2022) finetunes LLaMA (Touvron et al., 2023) with 52K instruction-following data using the `text-davinci-003` variant of GPT-3.5. It shows that the resulting Alpaca model behaves similarly to GPT-3.5 on instruction-following evaluation suite. Peng et al. (2023) further improves the performance by using instructions from the GPT-4 model. However, all these works focuses on the general setting with general user instructions. To the best of our knowledge, we are the first work on distillation from GPT-3/4 that focuses on a particular task. Our focus on summarization makes us able to use smaller models while not losing too much performance.

## 7 Conclusion & Future Works

We propose INHERITSUMM by distilling knowledge from GPT-3.5 using its summary on a broad range of documents. Base model of INHERITSUMM with only 400M parameters exhibits versatile capability on zeroshot, fewshot, and fully supervised summarization, surpassing performance of previous small models and beats or rivals GPT-3.5 in the overall performance.

For future works, it would be interesting to investigate the performance when distilling from other variants of GPT, like TEXT-DAVINCI-003 or GPT-4. Another interesting direction is controllable summarization - by using proper instructions, INHERITSUMM can be further trained to generate customized summarizations with style or length constraints.

## 8 Limitations

One limitation of INHERITSUMM is that it is only distilled in English and therefore lacks the multilingual generalization capability in other languages. Another limitation is that the distillation process

---

[3]We combine ① and ② because they are quite similar in style and focus on the same set of input documents.

of INHERITSUMM is cost inefficient as it requires extensive API calls to GPT.

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

## A List of Prompts

Below we list the prompts that we use from PromptSource. `[doc]` stands for the input document.

```
[doc] === Write a summary of the text above :  Summary:
[doc] How would you rephrase that in a few words?  Rephrase:
My college roommate asked me what this article means:  [doc] So I
recapped it in layman's terms:
Summarize this document:  [doc] Summary:
[doc] === Given the above document, write one sentence to summarize:
Summary:
First, please read the article below.  [doc] Now, can you write me an
extremely short abstract for it?  An extremely short abstract:
[doc] TL;DR:
Can you write an outline of the following article in a few points?
Article:  [doc] Outline:
Summarise the article:  [doc] Summary:
In 2 or 3 sentences, what are the main points one should remember
from this news article?  Article:  [doc] Main points:
Could you please generate a TLDR (Too Long Didn't Read) summary of
the following news article?  Article:  [doc] TLDR summary:
Condense the article down to the essentials to present it in the form
of short cards in mobile news apps:  [doc] Essentials:
Sum the following article in brief:  [doc] Breifs:
Extract key points from the article based on which the stock market
could react:  [doc] Key points:
Summarize this document:  [doc] Summary:
[doc] Given the above document, write a summary.  Summary:
Summarize:  [doc] Summary:
[doc] To sum up this document:
Sum up the following document:  [doc] Summary:
What are the key points across these news articles:  Article:  [doc]
Key points:
Synthesize these documents into a single one:  - [doc] Summary:
I want to edit the following articles into a more concise summary:
Article:  [doc] Summary:
Write a summary of the following articles:  Document:  [doc] Summary:
I'm trying to distill these articles down into one:  Article:  [doc]
Summary:
```

The special "following" prompt is

```
"Follow the example(s) above and summarize the document below:
Document:  [doc] Summary:
```

The in-context examples are prepended to the prompt in the format of `Document:  [doc] Summary:  [sum]` where `[doc]` and `[sum]` are the document and summary for the in-context examples respectively.

## B Sampling ratio of tasks

We mix ① and ② as one task, and treat ③④⑤⑥ as individual tasks. They are mixed by the ratio of [0.45,0.1,0.15,0.15, 0.15].

## C Sample Outputs

We list outputs from INHERITSUMM and baselines, together with the groundtruth human summaries. We can see that the outputs from INHERITSUMM follow GPT-3.5 and are often of equal or better quality.

GPT-3.5 sometimes fails to follow the instructions, and INHERITSUMM avoids such problems.

| System | Summary |
|---|---|
| IS-Large-C Zeroshot | The Rod of Asclepius is a symbol associated with the Greek god Asclepius, who was associated with healing and medicinal arts in Greek mythology. The symbol consists of a snake and a staff, which were sometimes depicted separately in antiquity. The most famous temple of Asclepius was at Epidaurus in north-eastern Peloponnese. The Rod of Asclepius has been used as a logo by a number of organizations and services. |
| IS-Large-C Fewshot | The Rod of Asclepius is a symbol associated with the Greek god Asclepius, a deity associated with healing and medicinal arts in Greek mythology. The symbol consists of a snake wrapped around a staff, and is thought to represent the dual nature of the work of the physician, who deals with life and death, sickness and health. The Rod of Asclepius is used as a logo by a number of organizations and services. |
| GPT-3.5 Zeroshot | the Rod of Asclepius. |
| GPT-3.5 Fewshot | The Rod of Asclepius is a staff with a snake wrapped around it that is used as a symbol of healing. It is named after the Greek god Asclepius, who was associated with healing and medicinal arts in Greek mythology. The symbol has been used by a number of organizations and services that are related to healing. |
| Human | In Greek mythology, the Rod of Asclepius (,, sometimes also spelled Asklepios or Asclepius), also known as the Staff of Aesculapius and as the asklepian, is a serpent-entwined rod wielded by the Greek god Asclepius, a deity associated with healing and medicine. The symbol has continued to be used in modern times, where it is associated with medicine and health care, yet frequently confused with the staff of the god Hermes, the caduceus. Theories have been proposed about the Greek origin of the symbol and its implications. |
| IS-Large-C Zeroshot | Charlotte Louise Juliette de Monaco was the illegitimate daughter of Marie Juliette Louvet, a cabaret singer, and Louis, Hereditary Prince of Monaco and Duke of Valentinois. She was adopted by Louis in 1919 and became heir presumptive to the throne as Hereditary Princess when her grandfather died and her father inherited the princely crown in 1922. Charlotte married Count Pierre de Polignac in 1920, but the marriage was not a happy one and they divorced in 1933. In 1944, Charlotte renounced and ceded her rights to the throne to her son Rainier. She died in 1977 in Paris. |

| System | Summary |
|---|---|
| IS-Large-C Fewshot | Charlotte Louise Juliette de Monaco was the illegitimate daughter of Marie Juliette Louvet, a cabaret singer, and Louis, Hereditary Prince of Monaco and Duke of Valentinois, son and heir of Monaco's reigning monarch, Prince Albert I. In 1911, a law was passed recognising Charlotte as Louis's daughter, and declaring her to be a dynastic member of the sovereign family. In 1919, Louis adopted Charlotte in Paris, and she became heir presumptive to the throne as Hereditary Princess when her grandfather died and her father inherited the princely crown in 1922. Charlotte married Count Pierre de Polignac in 1920, but the marriage was not a happy one and the couple divorced in 1933. In 1944, Charlotte renounced and ceded her rights to the throne to her son Rainier, subject to the stipulation that he did not predecease her. Charlotte died in 1977 in Paris. |
| GPT-3.5 Zeroshot | Charlotte was an illegitimate child of the heir to the throne of Monaco. In order to legitimize her and prevent a succession crisis, she was married off to a Frenchman who took the Grimaldi name. The couple had two children before divorcing. Charlotte later moved to Paris, where she died. |
| GPT-3.5 Fewshot | Princess Charlotte of Monaco was the illegitimate daughter of Marie Juliette Louvet, a cabaret singer, and Louis, Hereditary Prince of Monaco and Duke of Valentinois, son and heir of Monaco's reigning monarch, Prince Albert I. Louis had no legitimate children or siblings, so Charlotte was adopted into the dynasty in order to prevent a succession crisis. She married Count Pierre de Polignac of Hennebont, Morbihan, Brittany, France, who took the surname Grimaldi and became a Prince of the Monegasque Royal Family. The couple had two children, but their marriage was not a happy one and they divorced in 1933. Princess Charlotte renounced her rights to the throne in 1944, and she died in 1977. |
| Human | Princess Charlotte of Monaco, Duchess of Valentinois (Charlotte Louise Juliette Grimaldi; 30 September 1898 - 16 November 1977), was the daughter of Louis II, Prince of Monaco, and the mother of Rainier III, Prince of Monaco. From 1922 until 1944, she was the Hereditary Princess of Monaco, heir presumptive to the throne. |
| IS-Large-C Zeroshot | The article is discussing a pedal assembly that is used to control the throttle of a vehicle. The assembly includes a mounting bracket, a guide structure, a carrier assembly, a drive assembly, a pedal assembly, a resistance assembly, and a generator. The pedal assembly is adjustable to accommodate operators of varying anatomical dimensions, and the resistance assembly provides damping for a non-adjustable pedal assembly. |

| System | Summary |
|---|---|
| IS-Large-C Fewshot | The invention accelerator pedal assembly 10 includes a mounting bracket 11, a guide structure 12, a carrier assembly 13, a drive assembly 14, a pedal assembly 16, a resistance assembly 18, and a generator means 20. The pedal assembly 10 is intended to allow efficient fore and aft movement of the pedal assembly to accommodate operators of varying anatomical dimension and is operative to generate an electronic or drive - by - wire signal in response to pivotal movement of the pedal assembly while retaining the same ergometric operation of the pedal irrespective of the position of adjustment of the pedal. |
| GPT-3.5 Zeroshot | The invention pedal assembly provides the desired adjustability, the desired ergometrics, and the desired hysteresis effect in a structure that is simple, inexpensive, and positive and reliable in operation. |
| GPT-3.5 Fewshot | The invention provides a pedal assembly for a motor vehicle that can be adjusted to accommodate operators of varying anatomical dimensions, that has constant ergometrics irrespective of the position of adjustment of the pedal structure, and that provides a desired hysteresis effect in any position of adjustment of the pedal structure. The pedal assembly is simple, inexpensive, and positive and reliable in operation. |
| Human | an electronic adjustable control pedal assembly for a motor vehicle including a carrier , a guide rod adapted to be secured to the dash panel of the vehicle and mounting the carrier for fore and aft movement along the guide rod , a power drive operative to move the carrier along the guide rod , a pedal arm pivotally mounted on the carrier , a potentiometer mounted on the carrier and operative to generate an output electrical signal proportioned to the extent of pivotal movement of the pedal arm , and a resistance assembly to provide feedback or feel to the operator . the resistance assembly includes an annular friction surface defined on the carrier , a sleeve positioned over the friction surface , and a coil spring encircling the sleeve and arranged to be tightened in response to pivotal movement of the pedal arm to squeeze the sleeve against the friction surface and generate a friction resistance force . the friction resistance force adds to the torsional resistance force of the spring during application of the pedal and subtracts from the torsional resistance force of the spring upon release of the pedal , whereby to create a hysteresis effect . |

Table 7: Example zeroshot and fewshot outputs from IS-Large-C and GPT-3.5, together with ground truth human summaries.