# OpenReview forum: "InheritSumm: A General, Versatile and Compact Summarizer by Distilling from GPT"
_EMNLP/2023/Conference — EMNLP 2023 Findings_

### Official Review · Reviewer_UWoN · 2023-07-31

**Soundness:** 3

**Excitement:**

3: Ambivalent: It has merits (e.g., it reports state-of-the-art results, the idea is nice), but there are key weaknesses (e.g., it describes incremental work), and it can significantly benefit from another round of revision. However, I won't object to accepting it if my co-reviewers champion it.

**Paper Topic And Main Contributions:**

The paper introduces INHERITSUMM, a distilled, versatile model from GPT-3.5, addressing the challenges of large model fine-tuning costs while retaining robust zeroshot and fewshot summarization performance. Experimental results highlight its prowess.

**Reasons To Accept:**

1. The paper introduces INHERITSUMM, a versatile, compact summarization model distilled from GPT-3.5, displaying strong zeroshot and fewshot summarization potential similar to GPT-3.5 while being more compact.
2. A commendable effort was made to gather a large and diverse summarization dataset.
3. The use of ROUGE scores in experiments underscores the model's effectiveness.

**Reasons To Reject:**

1. While the abstract mentions the potential shortcomings of ROUGE scores, the experiments exclusively rely on them. This makes it uncertain if InheritSumm genuinely surpasses the baselines. The absence of case studies also limits insights into the model's performance.
2. The paper doesn't specify if the dataset or the model will be shared, and the lack of content in the appendix raises concerns about reproducibility, crucial for empirical research.

**Reproducibility:**

2: Would be hard pressed to reproduce the results. The contribution depends on data that are simply not available outside the author's institution or consortium; not enough details are provided.

**Reviewer Confidence:**

4: Quite sure. I tried to check the important points carefully. It's unlikely, though conceivable, that I missed something that should affect my ratings.

---

> ### Author Rebuttal · Authors · 2023-08-29
>
> Thank you for the insightful comments. Please see our response to your concerns below.
>
> 1）**Validity of the results**: While we are not able to include extra crowdsourced evaluations due to time and budget constraints, we will include extra case studies in the final version. Below are some examples. We can see that the outputs from InheritSumm follow GPT-3.5 and are often of equal or better quality. GPT sometimes fails to follow the instructions, and InheritSumm avoids such problems.
>
> 2）**Reproducibility**: We will share our data and model upon acceptance of the paper.
>
> Please let us know if you have further questions! Thank you.
>
> # Case Study
>
>
> | System | Summary |
> | --- | --- |
> | IS-Large-C Zeroshot | The Rod of Asclepius is a symbol associated with the Greek god Asclepius, who was associated with healing and medicinal arts in Greek mythology. The symbol consists of a snake and a staff, which were sometimes depicted separately in antiquity. The most famous temple of Asclepius was at Epidaurus in north-eastern Peloponnese. The Rod of Asclepius has been used as a logo by a number of organizations and services. |
> | IS-Large-C Fewshot | The Rod of Asclepius is a symbol associated with the Greek god Asclepius, a deity associated with healing and medicinal arts in Greek mythology. The symbol consists of a snake wrapped around a staff, and is thought to represent the dual nature of the work of the physician, who deals with life and death, sickness and health. The Rod of Asclepius is used as a logo by a number of organizations and services. |
> | GPT-3.5 Zeroshot | the Rod of Asclepius. |
> | GPT-3.5 Fewshot | The Rod of Asclepius is a staff with a snake wrapped around it that is used as a symbol of healing. It is named after the Greek god Asclepius, who was associated with healing and medicinal arts in Greek mythology. The symbol has been used by a number of organizations and services that are related to healing. |
> | Human | In Greek mythology, the Rod of Asclepius (,, sometimes also spelled Asklepios or Asclepius), also known as the Staff of Aesculapius and as the asklepian, is a serpent-entwined rod wielded by the Greek god Asclepius, a deity associated with healing and medicine. The symbol has continued to be used in modern times, where it is associated with medicine and health care, yet frequently confused with the staff of the god Hermes, the caduceus. Theories have been proposed about the Greek origin of the symbol and its implications. |
>
>
>
>
>
> | System | Summary |
> | --- | --- |
> | IS-Large-C Zeroshot | Charlotte Louise Juliette de Monaco was the illegitimate daughter of Marie Juliette Louvet, a cabaret singer, and Louis, Hereditary Prince of Monaco and Duke of Valentinois. She was adopted by Louis in 1919 and became heir presumptive to the throne as Hereditary Princess when her grandfather died and her father inherited the princely crown in 1922. Charlotte married Count Pierre de Polignac in 1920, but the marriage was not a happy one and they divorced in 1933. In 1944, Charlotte renounced and ceded her rights to the throne to her son Rainier. She died in 1977 in Paris. |
> | IS-Large-C Fewshot | Charlotte Louise Juliette de Monaco was the illegitimate daughter of Marie Juliette Louvet, a cabaret singer, and Louis, Hereditary Prince of Monaco and Duke of Valentinois, son and heir of Monaco's reigning monarch, Prince Albert I. In 1911, a law was passed recognising Charlotte as Louis's daughter, and declaring her to be a dynastic member of the sovereign family. In 1919, Louis adopted Charlotte in Paris, and she became heir presumptive to the throne as Hereditary Princess when her grandfather died and her father inherited the princely crown in 1922. Charlotte married Count Pierre de Polignac in 1920, but the marriage was not a happy one and the couple divorced in 1933. In 1944, Charlotte renounced and ceded her rights to the throne to her son Rainier, subject to the stipulation that he did not predecease her. Charlotte died in 1977 in Paris. |
> | GPT-3.5 Zeroshot | Charlotte was an illegitimate child of the heir to the throne of Monaco. In order to legitimize her and prevent a succession crisis, she was married off to a Frenchman who took the Grimaldi name. The couple had two children before divorcing. Charlotte later moved to Paris, where she died. |
> | GPT-3.5 Fewshot | Princess Charlotte of Monaco was the illegitimate daughter of Marie Juliette Louvet, a cabaret singer, and Louis, Hereditary Prince of Monaco and Duke of Valentinois, son and heir of Monaco's reigning monarch, Prince Albert I. Louis had no legitimate children or siblings, so Charlotte was adopted into the dynasty in order to prevent a succession crisis. She married Count Pierre de Polignac of Hennebont, Morbihan, Brittany, France, who took the surname Grimaldi and became a Prince of the Monegasque Royal Family. The couple had two children, but their marriage was not a happy one and they divorced in 1933. Princess Charlotte renounced her rights to the throne in 1944, and she died in 1977. |
> | Human | Princess Charlotte of Monaco, Duchess of Valentinois (Charlotte Louise Juliette Grimaldi; 30 September 1898 - 16 November 1977), was the daughter of Louis II, Prince of Monaco, and the mother of Rainier III, Prince of Monaco. From 1922 until 1944, she was the Hereditary Princess of Monaco, heir presumptive to the throne. |
>
>
>
>
>
>
> | System | Summary |
> | --- | --- |
> | IS-Large-C Zeroshot | The article is discussing a pedal assembly that is used to control the throttle of a vehicle. The assembly includes a mounting bracket, a guide structure, a carrier assembly, a drive assembly, a pedal assembly, a resistance assembly, and a generator. The pedal assembly is adjustable to accommodate operators of varying anatomical dimensions, and the resistance assembly provides damping for a non-adjustable pedal assembly. |
> | IS-Large-C Fewshot | The invention accelerator pedal assembly 10 includes a mounting bracket 11, a guide structure 12, a carrier assembly 13, a drive assembly 14, a pedal assembly 16, a resistance assembly 18, and a generator means 20. The pedal assembly 10 is intended to allow efficient fore and aft movement of the pedal assembly to accommodate operators of varying anatomical dimension and is operative to generate an electronic or drive - by - wire signal in response to pivotal movement of the pedal assembly while retaining the same ergometric operation of the pedal irrespective of the position of adjustment of the pedal. |
> | GPT-3.5 Zeroshot | The invention pedal assembly provides the desired adjustability, the desired ergometrics, and the desired hysteresis effect in a structure that is simple, inexpensive, and positive and reliable in operation. |
> | GPT-3.5 Fewshot | The invention provides a pedal assembly for a motor vehicle that can be adjusted to accommodate operators of varying anatomical dimensions, that has constant ergometrics irrespective of the position of adjustment of the pedal structure, and that provides a desired hysteresis effect in any position of adjustment of the pedal structure. The pedal assembly is simple, inexpensive, and positive and reliable in operation. |
> | Human | an electronic adjustable control pedal assembly for a motor vehicle including a carrier , a guide rod adapted to be secured to the dash panel of the vehicle and mounting the carrier for fore and aft movement along the guide rod , a power drive operative to move the carrier along the guide rod , a pedal arm pivotally mounted on the carrier , a potentiometer mounted on the carrier and operative to generate an output electrical signal proportioned to the extent of pivotal movement of the pedal arm , and a resistance assembly to provide feedback or feel to the operator . the resistance assembly includes an annular friction surface defined on the carrier , a sleeve positioned over the friction surface , and a coil spring encircling the sleeve and arranged to be tightened in response to pivotal movement of the pedal arm to squeeze the sleeve against the friction surface and generate a friction resistance force . the friction resistance force adds to the torsional resistance force of the spring during application of the pedal and subtracts from the torsional resistance force of the spring upon release of the pedal , whereby to create a hysteresis effect . |

---

### Official Review · Reviewer_mfxh · 2023-08-04

**Soundness:** 4

**Excitement:**

4: Strong: This paper deepens the understanding of some phenomenon or lowers the barriers to an existing research direction.

**Missing References:**

Not missing since it's concurrent work, but in case you missed it: On Learning to Summarize with Large Language Models as References


**Paper Topic And Main Contributions:**


The paper proposes a model distilled from GPT3.5 (text-davinci-002) on their curated GPT-SUMM dataset. The dataset consists of documents from the Pile corpus and supervised dataset and summaries are generated in several settings, including with/without in-context examples. The paper examines shortening demonstration documents to allow for more examples in few-shot learning. A ZCode++ model is trained on this data and compared in both zero and few-shot summarization settings.

**Questions For The Authors:**

1. Could you discuss more the effect of the choice of demonstration examples on model performance?
2. Could you discuss some of the qualitative differences in your model output across tasks and datasets?
3. The reader can estimate it, but if allowed, can you state the cost for the API calls to create the dataset?

**Reasons To Accept:**

1. Both the dataset and its methodology will be useful for future work and can inspire task-specific distillation studies.
2. The performance compares strongly against previous small models and GPT3.5

**Reasons To Reject:**

1. Analysis of model outputs would be helpful to better contextualize model performance. Only ROUGE is reported and discussed, without mention of other aspects such as factual consistency.

**Reproducibility:**

3: Could reproduce the results with some difficulty. The settings of parameters are underspecified or subjectively determined; the training/evaluation data are not widely available.

**Reviewer Confidence:**

4: Quite sure. I tried to check the important points carefully. It's unlikely, though conceivable, that I missed something that should affect my ratings.

---

> ### Author Rebuttal · Authors · 2023-08-29
>
> Thank you for the insightful comments. Please see our response to your questions below.
>
> 1）**Effect of the choice of demonstration examples**: We deliberately make the model robust to different types of demonstration examples by varying the demonstration example types when collecting the GPTSumm dataset (see Table 2). As a result, the InheritSumm model can take in a variety of demonstration examples not necessarily in the same style as the input document, and follow the demonstration to produce a new summary in the desired style. We also did an ablation study in Figure 2 on the number of demonstration examples. Generally, more demonstration examples help, and our succinct demonstration method is critical to generating high-quality summaries.
>
> 2）**Qualitative analysis**: We will include additional case studies in our final version. Below are some example outputs. The outputs from InheritSumm follow the pattern of GPT-3.5 and are often of equal or better quality. GPT sometimes fails to follow the instructions, and InheritSumm avoids such problems.
>
> 3）**Cost estimate**: We cannot reveal the exact cost due to confidentiality reasons, but we can give an estimate based on the public pricing. Our full dataset contains about 6.4M summaries generated by GPT3 (excluding existing datasets in part 6 in Table 2) In total there are 13 billion total input tokens including demonstration, and 1.6 billion output tokens for summary. This costs about $23k US dollars according to the current pricing of GPT-3 API.
>
> # Case Study
>
>
> | System | Summary |
> | --- | --- |
> | IS-Large-C Zeroshot | The Rod of Asclepius is a symbol associated with the Greek god Asclepius, who was associated with healing and medicinal arts in Greek mythology. The symbol consists of a snake and a staff, which were sometimes depicted separately in antiquity. The most famous temple of Asclepius was at Epidaurus in north-eastern Peloponnese. The Rod of Asclepius has been used as a logo by a number of organizations and services. |
> | IS-Large-C Fewshot | The Rod of Asclepius is a symbol associated with the Greek god Asclepius, a deity associated with healing and medicinal arts in Greek mythology. The symbol consists of a snake wrapped around a staff, and is thought to represent the dual nature of the work of the physician, who deals with life and death, sickness and health. The Rod of Asclepius is used as a logo by a number of organizations and services. |
> | GPT-3.5 Zeroshot | the Rod of Asclepius. |
> | GPT-3.5 Fewshot | The Rod of Asclepius is a staff with a snake wrapped around it that is used as a symbol of healing. It is named after the Greek god Asclepius, who was associated with healing and medicinal arts in Greek mythology. The symbol has been used by a number of organizations and services that are related to healing. |
> | Human | In Greek mythology, the Rod of Asclepius (,, sometimes also spelled Asklepios or Asclepius), also known as the Staff of Aesculapius and as the asklepian, is a serpent-entwined rod wielded by the Greek god Asclepius, a deity associated with healing and medicine. The symbol has continued to be used in modern times, where it is associated with medicine and health care, yet frequently confused with the staff of the god Hermes, the caduceus. Theories have been proposed about the Greek origin of the symbol and its implications. |
>
> | System | Summary |
> | --- | --- |
> | IS-Large-C Zeroshot | Charlotte Louise Juliette de Monaco was the illegitimate daughter of Marie Juliette Louvet, a cabaret singer, and Louis, Hereditary Prince of Monaco and Duke of Valentinois. She was adopted by Louis in 1919 and became heir presumptive to the throne as Hereditary Princess when her grandfather died and her father inherited the princely crown in 1922. Charlotte married Count Pierre de Polignac in 1920, but the marriage was not a happy one and they divorced in 1933. In 1944, Charlotte renounced and ceded her rights to the throne to her son Rainier. She died in 1977 in Paris. |
> | IS-Large-C Fewshot | Charlotte Louise Juliette de Monaco was the illegitimate daughter of Marie Juliette Louvet, a cabaret singer, and Louis, Hereditary Prince of Monaco and Duke of Valentinois, son and heir of Monaco's reigning monarch, Prince Albert I. In 1911, a law was passed recognising Charlotte as Louis's daughter, and declaring her to be a dynastic member of the sovereign family. In 1919, Louis adopted Charlotte in Paris, and she became heir presumptive to the throne as Hereditary Princess when her grandfather died and her father inherited the princely crown in 1922. Charlotte married Count Pierre de Polignac in 1920, but the marriage was not a happy one and the couple divorced in 1933. In 1944, Charlotte renounced and ceded her rights to the throne to her son Rainier, subject to the stipulation that he did not predecease her. Charlotte died in 1977 in Paris. |
> | GPT-3.5 Zeroshot | Charlotte was an illegitimate child of the heir to the throne of Monaco. In order to legitimize her and prevent a succession crisis, she was married off to a Frenchman who took the Grimaldi name. The couple had two children before divorcing. Charlotte later moved to Paris, where she died. |
> | GPT-3.5 Fewshot | Princess Charlotte of Monaco was the illegitimate daughter of Marie Juliette Louvet, a cabaret singer, and Louis, Hereditary Prince of Monaco and Duke of Valentinois, son and heir of Monaco's reigning monarch, Prince Albert I. Louis had no legitimate children or siblings, so Charlotte was adopted into the dynasty in order to prevent a succession crisis. She married Count Pierre de Polignac of Hennebont, Morbihan, Brittany, France, who took the surname Grimaldi and became a Prince of the Monegasque Royal Family. The couple had two children, but their marriage was not a happy one and they divorced in 1933. Princess Charlotte renounced her rights to the throne in 1944, and she died in 1977. |
> | Human | Princess Charlotte of Monaco, Duchess of Valentinois (Charlotte Louise Juliette Grimaldi; 30 September 1898 - 16 November 1977), was the daughter of Louis II, Prince of Monaco, and the mother of Rainier III, Prince of Monaco. From 1922 until 1944, she was the Hereditary Princess of Monaco, heir presumptive to the throne. |
>
> | System | Summary |
> | --- | --- |
> | IS-Large-C Zeroshot | The article is discussing a pedal assembly that is used to control the throttle of a vehicle. The assembly includes a mounting bracket, a guide structure, a carrier assembly, a drive assembly, a pedal assembly, a resistance assembly, and a generator. The pedal assembly is adjustable to accommodate operators of varying anatomical dimensions, and the resistance assembly provides damping for a non-adjustable pedal assembly. |
> | IS-Large-C Fewshot | The invention accelerator pedal assembly 10 includes a mounting bracket 11, a guide structure 12, a carrier assembly 13, a drive assembly 14, a pedal assembly 16, a resistance assembly 18, and a generator means 20. The pedal assembly 10 is intended to allow efficient fore and aft movement of the pedal assembly to accommodate operators of varying anatomical dimension and is operative to generate an electronic or drive - by - wire signal in response to pivotal movement of the pedal assembly while retaining the same ergometric operation of the pedal irrespective of the position of adjustment of the pedal. |
> | GPT-3.5 Zeroshot | The invention pedal assembly provides the desired adjustability, the desired ergometrics, and the desired hysteresis effect in a structure that is simple, inexpensive, and positive and reliable in operation. |
> | GPT-3.5 Fewshot | The invention provides a pedal assembly for a motor vehicle that can be adjusted to accommodate operators of varying anatomical dimensions, that has constant ergometrics irrespective of the position of adjustment of the pedal structure, and that provides a desired hysteresis effect in any position of adjustment of the pedal structure. The pedal assembly is simple, inexpensive, and positive and reliable in operation. |
> | Human | an electronic adjustable control pedal assembly for a motor vehicle including a carrier , a guide rod adapted to be secured to the dash panel of the vehicle and mounting the carrier for fore and aft movement along the guide rod , a power drive operative to move the carrier along the guide rod , a pedal arm pivotally mounted on the carrier , a potentiometer mounted on the carrier and operative to generate an output electrical signal proportioned to the extent of pivotal movement of the pedal arm , and a resistance assembly to provide feedback or feel to the operator . the resistance assembly includes an annular friction surface defined on the carrier , a sleeve positioned over the friction surface , and a coil spring encircling the sleeve and arranged to be tightened in response to pivotal movement of the pedal arm to squeeze the sleeve against the friction surface and generate a friction resistance force . the friction resistance force adds to the torsional resistance force of the spring during application of the pedal and subtracts from the torsional resistance force of the spring upon release of the pedal , whereby to create a hysteresis effect . |

---

### Official Review · Reviewer_FFF7 · 2023-08-06

**Soundness:** 3

**Excitement:**

3: Ambivalent: It has merits (e.g., it reports state-of-the-art results, the idea is nice), but there are key weaknesses (e.g., it describes incremental work), and it can significantly benefit from another round of revision. However, I won't object to accepting it if my co-reviewers champion it.

**Paper Topic And Main Contributions:**

Although the paper is titled InheritSumm, it is actually about a new dataset GPTSUMM that the authors created. They showed that if an existing model such as  ZCode++ is trained on this dataset, the resulting model's output for the summarization task is comparable to that of GPT-3.5.

**Questions For The Authors:**

Question A:
Why did you prefer ROUGE-2 instead ROUGE-L?

**Reasons To Accept:**

The new GPTSUMM dataset will be useful for those in community working on summarization. The model trained using this dataset produces summarization output comparable to GPT-3.5.

**Reasons To Reject:**

For me, this paper contains enough work to be accepted as a short paper but not enough for a long paper. The paper is titled InheritSumm which is misleading. The main contribution of this paper is the GPTSUMM dataset. For training the InheritSumm models the authors simply applied exiting  ZCode++ on the GPTSUMM dataset.

I would have still considered it about InheritSumm if the author included sufficient analysis of the InheritSumm results. For example, as shown in Table 5, there is basically no improvement between InheritSumm-base and InheritSumm-large models. In fact,  InheritSumm-base performs slightly better than InheritSumm-base-large. The authors mentioned they use ZCode++ for training both models using their dataset. But they did explain the peculiarity in the results.

**Reproducibility:**

3: Could reproduce the results with some difficulty. The settings of parameters are underspecified or subjectively determined; the training/evaluation data are not widely available.

**Reviewer Confidence:**

3: Pretty sure, but there's a chance I missed something. Although I have a good feel for this area in general, I did not carefully check the paper's details, e.g., the math, experimental design, or novelty.

**Typos Grammar Style And Presentation Improvements:**

Page 1

in the the way => in the way

---

> ### Author Rebuttal · Authors · 2023-08-29
>
> Thank you for the insightful comments, and we are glad that you also agree on GPTSumm’s great value in developing universal summarization models. Please see our response to your concerns below.
>
> 1. **Not enough contribution for the InheritSumm model**: We respectfully disagree with this argument. While the GPTSumm dataset is indeed a significant contribution, InheritSumm serves as a versatile, compact model capable of summarizing a wide range of document styles. The model's release is pivotal for the research community and opens avenues for future work. Method-wise, we also propose to use succinct demonstrations for distilling from GPT-3 on the summarization task (Sec 3.4). Given that summarization tasks often involve long inputs, this distillation method is particularly crucial. To the best of our knowledge, the method is novel for summarization distillation.
>
> 2. **Comparison between base and large models**: We acknowledge the observation made in Table 5 regarding the performance similarities between the base and large models. Our hypothesis is that this may be attributed to the inherent characteristics of the ZCode++ models. In our experiments, the base model has shown competitive performance across various tasks, not just in summarization. Nevertheless, the fact that the base model performs on par with the large while outperforming other models is a benign phenomenon - It suggests that excellent performance can be achieved with a more resource-efficient model.
>
> 3. **Question A: ROUGE-2 vs ROUGE-L**: We choose ROUGE-2 over other metrics because previous studies [1] find that ROUGE-2 has a better correlation with human judgments on meta-eval benchmarks like SummEval, and SummEval has a similar composition of datasets as our testing benchmark. We will make this point clear in the final version.
>
> We hope this addresses your concerns and questions adequately. Should you have any further queries, please do not hesitate to ask.
> Thank you for your time and consideration.
>
> [1] Zhong, Ming, et al. "Towards a unified multi-dimensional evaluator for text generation." arXiv preprint arXiv:2210.07197 (2022).

---

### Meta-Review · Area_Chair_YXfQ · 2023-09-17

**Recommendation:** 4

**Metareview:**

The paper presents a large curated dataset GPTSumm, collected using documents from Pile as the source and GPT3.5 summaries. The paper studies the performance of a summarization model distilled from this dataset. Both these are valuable contributions for future work on model distillation for summarization. The evaluation of the model is missing any human evaluation or discussion/analysis of factuality of generated summaries (standard now for summarization papers); these will strengthen the claims in the paper.

---

### Decision · Program_Chairs · 2023-10-07

**Decision:**

Accept-Findings

**Comment:**

The paper presents a large curated dataset GPTSumm, collected using documents from Pile as the source and GPT3.5 summaries. The paper studies the performance of a summarization model distilled from this dataset. Both these are valuable contributions for future work on model distillation for summarization. The evaluation of the model is missing any human evaluation or discussion/analysis of factuality of generated summaries (standard now for summarization papers); these will strengthen the claims in the paper.